# Temporal Changes in the Composition of Beached Holopelagic *Sargassum* spp. along the Northwestern Coast of Cuba



Eduardo Gabriel Torres-Conde [1,2,*], Brigitta I. van Tussenbroek [1], Rosa E. Rodríguez-Martínez [1] and Beatriz Martínez-Daranas [2]

1 Instituto de Ciencias del Mar y Limnología-UNAM, Unidad Académica de Sistemas Arrecifales, Prol. Av. Niños Héroes S/N, Puerto Morelos C.P. 77580, Q. Roo, Mexico; vantuss@cmarl.unam.mx (B.I.v.T.); rosaer@cmarl.unam.mx (R.E.R.-M.)

2 Centro de Investigaciones Marinas, Universidad de la Habana, Calle 16 No. 114, Playa, La Habana C.P. 11300, Cuba; bmdaranas@gmail.com

* Correspondence: etorresconde2@gmail.com

**Abstract:** Since 2011, the distribution, abundance, and composition of holopelagic *Sargassum* spp. (sargasso) have changed by the emergence of the Great Atlantic Sargasso Belt (GASB) in the northern tropical Atlantic. We expected that the north of the Cuban coast would receive sargasso from both the original Sargasso Sea and the GASB. We systematically monitored six beaches on the NW coast of Cuba to assess changes in sargasso composition from June 2019 to June 2021. During landing months, mean Sargasso wet biomass was at 1.54 kg/m$^2$ (SE: 0.7), which was considerably lower than the sargasso on the Atlantic coasts directly impacted by GASB. Eleven out of 13 landings occurred in the autumn-winter seasons 2019–2020 and 2020–2021, with a dominance of *S. natans* I (accounting for 41–63% of total biomass), followed by *S. fluitans* III (25–36%) and *S. natans* VIII (12–31%). This composition is similar to those observed on the Sargasso Sea. During this season, dominant winds ($\geq$14 km/h) came from northern (N), eastern (E), and east-northeastern (ENE) directions. In May and August 2020 (spring-summer season), *S. fluitans* III dominated (52–56%), followed by *S. natans* VIII (33–43%) and *S. natans* I (5–12%). This composition is similar to those observed on GASB-impacted Atlantic coasts in the spring-summer seasons (April to September). During this season, dominant winds ($\geq$20 km/h) came from eastern (E) and east-northeastern (ENE) directions. Thus, the NW Cuba's morphotype composition suggests that landings have different origin sources depending on season and specific meteorological and oceanographic conditions.

**Keywords:** beach cast; macroalgal bloom; morphotype; sargasso; northwestern tropical Atlantic

## 1. Introduction

Holopelagic *Sargassum* species (*S. fluitans* and *S. natans*; sargasso from hereon) are free-floating brown macroalgae that populate the open ocean transported by winds and currents [1,2]. These macroalgae present elongated branching structures adorned with small bladder-like appendages that promote buoyancy [3,4]. They have existed in the northern Atlantic for at least centuries, concentrated in the Sargasso Sea, as evidenced by historical records from early explorations and maritime history [3]. Sargasso has not been contained absolutely in the Sargasso Sea, as evidenced by the Sargasso Loop System [5], which has usually caused minor landings on the coasts of NW Caribbean islands and mainland and the Gulf of Mexico (Figure 1). Pelagic masses of these species support a biodiverse community by providing sustenance, refuge, and protection to many organisms, such as fish, sea turtles, and invertebrates [6].

The distribution and abundance of sargasso in the Atlantic Ocean have changed since 2011, with the formation of the Great Atlantic Sargasso Belt (GASB) in the northern tropical Atlantic [7,8] (Figure 1). The proliferation of these macroalgae in the GASB has presented

numerous challenges, such as excessive accumulations detrimentally impacting coastal ecosystems and communities reliant on tourism [9–11].

Sargasso exhibits distinct morphological forms, first defined by Parr in 1939. Present-day pelagic masses contain a mixture of distinct forms with the dominance of *S. fluitans* III, *S. natans* I, and *S. natans* VIII, and their relative abundance varies across regions, seasons, and years [12,13]. Studies have revealed clear genetic differentiation among the species and morphotypes [14,15]. They also can differ in resident fauna [13,16], chemical composition [17–23], and growth rates [24–26].

Various studies have quantified changes in the composition of beached sargasso on the Atlantic coasts, which is more feasible than studying the pelagic masses in the open ocean [21,27–30]. We conducted systematic beach monitoring from 2019 to 2021 to quantify changes in the composition of holopelagic sargasso on six beaches along the NW coast of Cuba (Figure 1). This coast likely receives sargasso from the Sargasso Sea, the Gulf of Mexico, and GASB (Figure 1), which may be reflected in the specific composition.

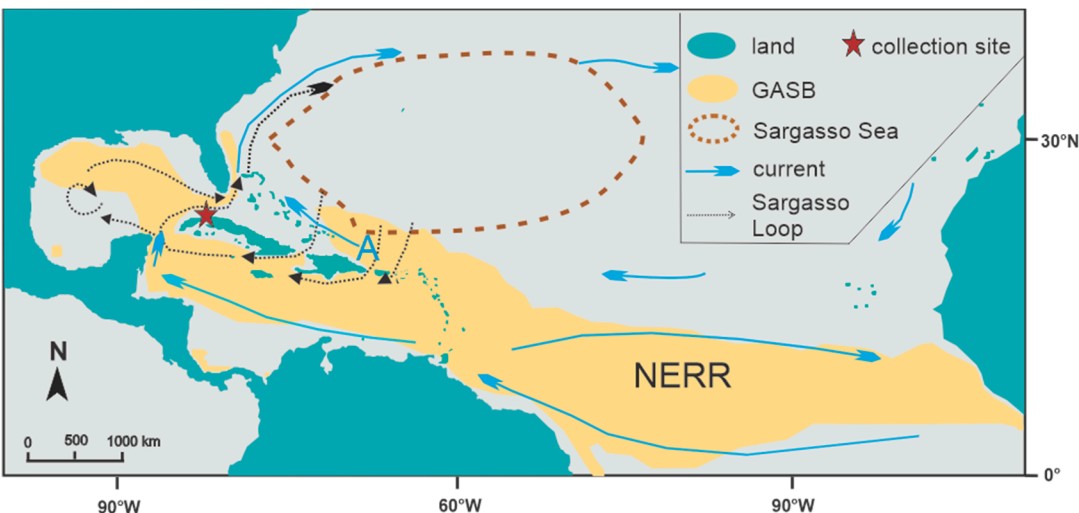

**Figure 1.** Potential distribution of holopelagic *Sargassum* spp. in the northern tropical and subtropical Atlantic (GASB after [31]; Sargasso Sea based on https://oceanfdn.org/sargasso-sea/, accessed on 13 August 2023), with approximate significant currents (A indicating the Antilles current) and the sargasso loop (after [5]). GASB Great Atlantic Sargasso belt, NERR North Equatorial Recirculation region.

## 2. Materials and Methods

### 2.1. Beach Surveys

The study was conducted on six beaches on the NW coast of Cuba (Table 1), exposed to prevailing strong winds from the north during the winter and easterlies throughout the rest of the year [32]. During the study period, no hurricanes passed near the study area. A 50 m measuring tape was positioned parallel to the shoreline in the intertidal zone, and five points were selected randomly along the tape, replicating the process along the whole length of each beach (see Table 1). Sargasso was sampled within a 1 m$^2$ quadrant at each point. All identifiable fresh, golden-colored sargasso was collected, taken to the laboratory, and separated by morphotypes following [3,12]. The wet weight of each morphotype per sample was measured using an OHAUS spring scale after removing excess water with a towel. Surveys were conducted monthly from June 2019 to June 2021.

**Table 1.** Characteristics of the beaches surveyed along the NW Cuban coastline. N number of samples collected monthly.

| Beach | Coordinates | Length of the Beach (m) | Supralittoral Composition | N | Mean (±SD) Biomass (wet kg m$^{-2}$) |
|---|---|---|---|---|---|
| Cojímar | 23°09′47″ N, 82°17′38″ W | ~370 | Sandy | 25 | 1.12 ± 0.4 |
| Bacuranao | 23°10′36″ N, 82°14′30″ W | ~319 | Sandy | 25 | 1.35 ± 0.8 |
| Tarará | 23°10′50″ N, 82°12′11″ W | ~500 | Sandy | 35 | 2.44 ± 1.1 |
| Mégano | 23°10′45″ N 82°11′53″ W | ~500 | Sandy | 35 | 2.25 ± 1.2 |
| Santa María | 23°10′41″ N 82°11′42″ W | ~3000 | Sandy | 35 | 2.45 ± 1.4 |
| Paseo Marítimo | 23°6′49″ N 82°26′24″ W | ~350 | Rocky | 25 | 1.18 ± 0.75 |

*2.2. Data Analysis*

The monthly mean biomass and relative abundance of the sargasso morphotypes were calculated for each beach. Possible differences in the biomass during the landing months were tested with a one-way ANOVA. A Pearson Correlation test assessed the possible relation between biomass and wind speed or wave height. Data on wind speed and direction, as well as wave height and direction, were collected from the Windguru website (https://www.windguru/cz/ accessed on 14 August 2023) for the day before sampling. This information was obtained from the GFS (Global Forecast System) model with a resolution of 27 km and was measured four times per day (00 UTC, 06 UTC, 12 UTC, and 18 UTC). A significance level of 0.05 was used for all tests. All analyses were performed in R [33]. The data complied with the assumptions of homogeneity of variances (Levene test) and normality (Shapiro–Wilk test).

**3. Results**

Sargasso landed in NW Cuba in 13 out of the 25 monthly surveys conducted from June 2019 to June 2021 (Figure 2). No differences in biomass were found during the landing sampling dates (df = 13, F = 1.34, *p* = 0.25) (Table 2). The mean sargasso wet biomass during the landing months was 1.56 kg m$^2$ (SE: 0.4) and was composed of the three commonly occurring morphotypes that have been reported in the Caribbean (Supplementary Table S1).

The morphotype composition of beached sargasso in NW Cuba was relatively consistent during the autumn-winter season of 2019–2020 and 2020–2021, when the dominant morphotype was *S. natans* I (% biomass range: 41–63%), followed by *S. fluitans* III (25–36%) and *S. natans* VIII (12–31%). The composition was atypical in the study area in May and August 2020 (spring-summer season), when *S. fluitans* III became dominant (52–56%), followed by *S. natans* VIII (33–43%) and *S. natans* I was rare (5–12%) (Figure 2).

The biomass of sargasso in NW Cuba was strongly correlated with wind speed (df = 23, *p* < 0.000, R = 0.81) (≥14 km/h) and moderately correlated with wave height (df = 23, *p* < 0.000, R = 0.67) (≥0.8 m), coming predominantly from the north, east and east-northeast directions (Table 2).

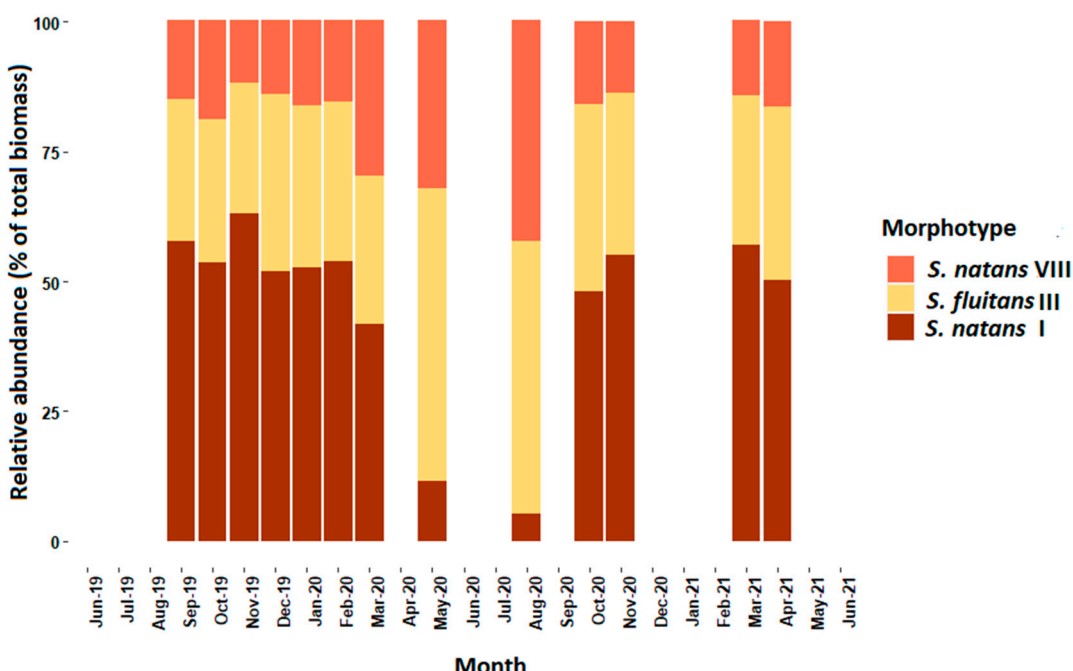

**Figure 2.** Relative abundance of each holopelagic *Sargassum* morphotype beached at the northwestern coast of Cuba. The absence of a bar indicates that no sargasso was found.

**Table 2.** Means (±SE) of sargasso wet biomass, wind speed (km/h), wave height (m), and direction of prevailing winds and waves during the sampling period (June 2019 to June 2021) in the northwestern coast of La Habana, Cuba. Data on meteorological variables was obtained for the day before sampling. nd: No wind or wave direction was detected due to weak winds.

| Sampling Date | Wet Biomass (kg m²) | Wind Speed (km/h) | Prevailing Wind Direction | Wave Height (m) | Prevailing Wave Direction |
|---|---|---|---|---|---|
| 20 June 2019 | 0 | 9 ± 1.1 | E | 0.4 ± 0.2 | E |
| 22 July 2019 | 0 | 12.2 ± 1.2 | SE | 0.7 ± 0.1 | E |
| 25 August 2019 | 0 | 9.26 ± 1.5 | E | 0.5 ± 0.1 | E |
| 21 September 2019 | 0.50 ± 0.16 | 18.5 ± 0.5 | ENE | 1.7 ± 0.3 | ENE |
| 25 October 2019 | 0.81 ± 0.21 | 18.13 ± 0.4 | E | 1.1 ± 0.3 | E |
| 22 November 2019 | 0.90 ± 0.35 | 14.5 ± 2.1 | ENE | 0.9 ± 0.2 | ENE |
| 22 December 2019 | 0.99 ± 0.34 | 21.5 ± 1.2 | ENE | 0.9 ± 0.3 | ENE |
| 22 January 2020 | 1.70 ± 0.46 | 18.75 ± 1.1 | N | 1.7 ± 0.3 | N |
| 23 February 2020 | 1.69 ± 0.25 | 19.5 ± 0.3 | NNE | 1.3 ± 0.2 | NNE |
| 21 March 2020 | 2.39 ± 0.24 | 18 ± 1.12 | E | 1.5 ± 0.3 | E |
| 22April 2020 | 0 | 8.7 ± 1.1 | nd | 0.4 ± 0.1 | W |
| 15 May 2020 | 0.90 ± 0.30 | 21.75 ± 2.3 | ENE | 2 ± 0 | ENE |
| 22 June 2020 | 0 | 9.5 ± 1.2 | E | 0.4 ± 0.1 | E |
| 20 July 2020 | 0 | 11 ± 2.1 | E | 0.85 ± 0.1 | E |
| 25 August 2020 | 2.17 ± 0.8 | 25.5 ± 2.3 | E | 2.5 ± 1.1 | E |
| 22 September 2020 | 0 | 5.25 ± 0.3 | nd | 0.45 ± 0.1 | N |
| 20 October 2020 | 1.63 ± 0.34 | 18.25 ± 2.1 | E | 0.8 ± 0.1 | ENE |
| 20 November 2020 | 1.83 ± 0.68 | 24.25 ± 1.7 | ENE | 2.5 ± 0.2 | ENE |
| 18 December 2020 | 0 | 6.25 ± 1.4 | NNW | 0.43 ± 0.1 | NNW |
| 24 January 2021 | 0 | 6 ± 0.8 | nd | 0.1 ± 0.01 | nd |
| 17 February 2021 | 0 | 7.25 ± 0.7 | S | 0.6 ± 0.5 | S |
| 13 March 2021 | 1.88 ± 0.62 | 19 ± 1.1 | ENE | 1.4 ± 1.3 | ENE |
| 24 April 2021 | 2.06 ± 0.51 | 14.74 ± 1.6 | ENE | 0.4 ± 0.2 | ENE |
| 16 May 2021 | 0 | 12 ± 1.1 | ENE | 0.8 ± 0.2 | ENE |
| 17 June 2021 | 0 | 6.25 ± 0.6 | nd | 0.23 ± 0.01 | nd |

## 4. Discussion

The mean wet biomass that landed on the NW Cuban coast was considerably lower than that reported for other regions affected by massive landings from the Great Atlantic Sargasso Belt (GSAB). For example, the mean wet biomass accumulation of stranded sargasso was reported at 98 kg m$^2$ at Atalaia Beach, Brazil, in the peak event of 2015 [34], at 42.0 kg m$^2$ (SE: 6.7) in the Mexican Caribbean in August 2018 [27], and at 95 kg m$^2$ in the Costa Rica coast in March-April 2019 [29]. The stranded biomass recorded in NW Cuba was more like that documented on the coast of Florida (4.0 kg m$^2$) in April 2019 [28]. According to [28], the geographical location of the Floridian and Gulf of Mexico coasts may favor lower landings of sargasso, which could extend to include the NW Cuba coasts. In addition, the low landing biomass remained relatively constant throughout the study period, which suggests that the NW Cuba coast receives low quantities of sargasso independently of the season and possible distinct origin source.

The dominance of *S. natans* I documented in NW Cuba ($\geq$45%) during most beaching events is consistent with that reported by [3] and [12–14] for the Sargasso Sea, where this morphotype is dominant, representing from 43–90% of the biomass. In contrast, it differs from reports documented in other sites affected by massive sargasso landings from the GASB, where *S. fluitans* III is usually the dominant morphotype [27,28,30]. For example, in the Mexican Caribbean, monthly surveys of beach cast sargasso from 2016 to 2020 showed that ~60% of the biomass in most of the monitoring months was accounted for by *S. fluitans* III, whereas *S. natans* I was less common, except for one landing event during March 2019 [27]. *S. fluitans* III was also dominant in samples from Florida and the Bahamas from 2019 to 2021 (50% and 55%, respectively) and Barbados from 2021 to 2022 (80%) [28]. The dominance of *S. natans* I in NW Cuba also differs from the composition found in 2014–2015 in the Tropical Atlantic, the Eastern Caribbean, and the Antilles Current, where *S. natans* VIII was more abundant (>85%) [12,13] also found low abundance of *S. natans* I in the Tropical Atlantic, the Greater Caribbean, the Gulf of Mexico, and the North Equatorial Recirculation Region (NERR).

The contrasting morphotype composition of landings in NW Cuba with time could indicate distinct origin sources, as the proximity to the sargasso source can influence the morphotype composition and the accumulation level of the landings [28]. Landings of sargasso in northern Cuba coasts have been historically associated with the Sargasso Sea and its proximity [35–39]. Various studies have documented low landings (0.73 $\pm$ 0.54 kg/m$^2$) in the winter season (October to March; sometimes extended from September to April) when winds had velocities of $\geq$10 km/h and wind and wave directions from the N, E, and ENE, associated with the presence of cold fronts and the influence of Continental Anticyclone [35–39]. In addition to currents, windage has been found to be relevant for sargasso transport [40], and in our study, landings were strongly correlated with wind speed $\geq$ 14 km/h and wind and wave directions mainly from N, E, and ENE being consistent during the 2019–2020 winter season but not as much in the 2020–2021 winter season. Probably, the occurrence of weak winds $\leq$ 8 km/h and wind and wave direction from NNW and S in November 2020, December 2020, and January 2021 did not promote landings in this winter season.

In both May and August of 2020, the NW Cuba coasts were dominated by *S. fluitans* III and *S. natans* VIII. During the spring-summer season (April to September), when GASB tends to be more abundant, coasts that commonly do not receive sargasso from the GASB may receive biomass that did not follow the main trajectories of the Caribbean, Caiman and Yucatan, Loop Currents, and Gulf Stream (Figure 1). Probably, sargasso from the GASB may have reached the NW Cuban coast in the summer of 2020 through similar transport ways. Satellite images did not show important sargasso biomass near Cuba's NW coast during the months when *S. fluitans* III was the dominant landing morphotype (https://optics.marine.usf.edu/projects/SaWS.html, accessed on 11 September 2023). However, small isolated and scattered rafts are not detected using satellites, and according to Ody et al. (2019) [41], small-size rafts (~30 cm) represented 14% of the rafts

observed in situ surveys in the central Tropical North Atlantic in the summer of 2017, and that they were more abundant in the northern part of the eastern Caribbean. Alternatively, sargasso from the GASB may have been transported to the NW coasts via the Antilles Current and trade winds; further modeling and tracking analysis may discern the possible routes of transport of sargasso onto the Cuban coasts. Thus, the NW Cuba's morphotype composition suggests that landings have different origin sources depending on season and specific meteorological and oceanographic conditions. In the winter, the NW Cuban coast likely received sargasso from the Sargasso Sea (*S. natans* I dominated), whereas in the summer months, sargasso likely came from the GASB (*S. fluitans* III dominated).

Landings of sargasso are irregular and unpredictable [42,43]. Climate change is causing anomalous temperatures, heatwaves, and abnormal meteorological conditions [44], while species/morphotypes of sargasso exhibit varying thermal tolerances and growth rates [24–26]. Consequently, the composition and abundance of beached and oceanic sargasso (in both the Sargasso Sea and GASB) are likely subject to ongoing changes. Before 2011, offshore surveys by [3] indicated that *S. fluitans* III and *S. natans* I were the dominant species in the Atlantic Ocean, with *S. natans* VIII being rare. Post-2011, various studies have documented a high dominance of *S. natans* VIII in the Atlantic Ocean, except in the Sargasso Sea where *S. natans* I has remained dominant, though its composition is increasingly mixed [12,13]. Ongoing monitoring of the specific composition of sargasso throughout the northern tropical Atlantic will aid in understanding the fluxes of sargasso in the northern tropical Atlantic.

**Supplementary Materials:** The following supporting information can be downloaded at: https://www.mdpi.com/article/10.3390/phycology3040027/s1, Table S1. Spatial variability of the holopelagic sargasso morphotypes biomass (Mean ± SD) during the landings months (N = 13) for the surveyed beaches.

**Author Contributions:** Conceptualization, E.G.T.-C. and B.M.-D.; methodology, E.G.T.-C. and B.M.-D.; formal analysis, E.G.T.-C.; investigation, E.G.T.-C., B.I.v.T. and R.E.R.-M.; resources, E.G.T.-C. and B.M.-D.; data curation, E.G.T.-C.; writing—original draft preparation, E.G.T.-C., B.I.v.T. and R.E.R.-M.; writing—review and editing, E.G.T.-C., B.I.v.T., R.E.R.-M. and B.M.-D.; visualization, E.G.T.-C., B.I.v.T. and R.E.R.-M.; supervision, E.G.T.-C., B.I.v.T., R.E.R.-M. and B.M.-D.; project administration, E.G.T.-C. and B.M.-D.; funding acquisition, E.G.T.-C. and B.M.-D. All authors have read and agreed to the published version of the manuscript.

**Funding:** This research was funded by FONCI (Project "Reciclado de nutrientes y carbón a partir de biomasa para fertilización orgánica de avanzada en la agricultura en Cuba eco-inteligente y climáticamente positiva").

**Institutional Review Board Statement:** Not applicable.

**Informed Consent Statement:** Not applicable.

**Data Availability Statement:** The data presented in this study are available on request from the corresponding author.

**Acknowledgments:** We thank all the people who in one way or another helped with the field data collection. Funding from FONCI (Project "Reciclado de nutrientes y carbón a partir de biomasa para fertilización orgánica de avanzada en la agricultura en Cuba eco-inteligente y climáticamente positiva") is acknowledged.

**Conflicts of Interest:** The authors declare no conflict of interest. The funder had no role in the study design, data collection, and interpretation or the decision to submit the work for publication.

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
