# Peer review of "Temporal Changes in the Composition of Beached Holopelagic Sargassum spp. along the Northwestern Coast of Cuba"

_phycology, doi:10.3390/phycology3040027_

Round 1
Reviewer 1 Report
The study presented here shows the variability of morphotypes composition of beached holopelagic Sargassum. A focus is done on the northwestern coast of Cuba.
The study is interesting as the authors' aim is to determine the origin of holopelagic Sargassum that washes up in their study area. For this aim, the authors give the composition of morphotypes measured in 6 sites in the northwestern coast of Cuba
After reading the manuscript, I recommend this manuscript for a publication in the Journal Phycology, as it gives interesting results. But before that, I propose a minor revision of the manuscript before it could be accepted for publication. I explain the various points for improvement below.
From a general point of view, results are clear, nevertheless, I have a question about the methodology and the obtained results.
Why did the authors select only sites on the west coast and not in the east?
Because if you follow the current and take account of episodic/abnormal events, the east coast could also be the site of beachings? Please explain
If I've understood correctly, Figure 2 and Table 1 show average values for the 6 sites studied... why did you average the sites?
Table 1. Because it would be interesting to know whether more strandings occur on a given site (sandy beaches, for example) or, on the contrary, whether these strandings are homogeneous whatever the site (rocky, sandy).
Figure 2. It would be interesting to know whether or not strandings are homogeneous depending on the site and the time of year.
Concerning the morphotypes composition, results showed a dominance of S. natans I in Autumn/Winter 2019/2020 while in Spring/Summer 2020, S. natans I decreased drastically and S. fluitans III dominated
What is the conclusion about these tendancies?
I wonder is the aim of the study is completed, as I do not find any hypotheses on the origin of Sargassum in the northwestern coast of Cuba
From which part of the Atlantic ocean are originated beached Sargassum in the northwestern part of Cuba? One or several origins?
In the discussion section, it will be interesting to add the article from Ody et al. (2019) which presented also (floating) biomasses estimation more South compared to the study presented here. These (drift) biomasses presented in Ody et al. (2019) are more important compared to the study here (beached biomasses) but comparable with those from Wang et al. (2019) who presented also drift biomasses. Hypotheses could be done also on these decrease of biomasses (from South to North, from drift to beachings).
Bibliographic references are good and relevant
Author Response
We thank the reviewers for their comments and suggestions. We have made all the suggested changes.
Reviewer #1:
- Why did the authors select only sites on the west coast and not in the east?
Because if you follow the current and take account of episodic/abnormal events, the East Coast could also be the site of beachings? Please explain
Reply: This is good observation, and ideally, we should have included the east coast in the study to obtain a more complete picture of the landings of sargasso. However, this was not possible for logistical reasons, due to insufficient funding.
2) If I've understood correctly, Figure 2 and Table 1 show average values for the 6 sites studied... why did you average the sites?
Table 1. Because it would be interesting to know whether more stranding occur on a given site (sandy beaches, for example) or, on the contrary, whether these stranding are homogeneous whatever the site (rocky, sandy).
Figure 2. It would be interesting to know whether or not stranding are homogeneous depending on the site and the time of year.
Reply: Thank you for this observation, and we agree that spatial variability is also interesting and we have now presented the information per site in supplementary Table 1. We do not present this information in the main text as the objective of our study was to determine general tendencies in biomass and relative abundance of morphotypes, but data per site may be of interest to some readers.
3) Concerning the morphotypes composition, results showed a dominance of S. natans I in Autumn/Winter 2019/2020 while in Spring/Summer 2020, S. natans I decreased drastically and S. fluitans III dominated
What is the conclusion about these tendencies?
Reply: We thought that we had explained the tendencies are in the manuscript in the following paragraph (lines 146-150):
“The contrasting morphotype composition of landings in NW Cuba with time could indicate distinct origin sources, as the proximity to the sargasso source can influence the morphotype composition and the accumulation level of the landings”.
The text was modified in lines 146-154 as follows:
The contrasting morphotype composition of landings in NW Cuba with time could indicate distinct origin sources, as the proximity to the sargasso source can influence the morphotype composition and the accumulation level of the landings [28]. Landings of sargasso in northern Cuba coasts have been historically associated with the Sargasso Sea and its proximity [35-39]. Various studies have documented low landings (0.73 ± 0.54 kg/m2) in the winter season (October to March; sometimes extended from September to April) when winds had velocities of ≥ 10 km/h and wind and wave directions were from the N, E, and ENE, associated with the presence of cold fronts and the influence of Continental Anticyclone [35-39].
However, likely we were not sufficiently explicit, and we have added the following text in lines 181-183.
“In the winter, the NW Cuban coast likely received sargasso from the (S. natans I dominated) Sargasso Sea, whereas in the summer months, sargasso likely came from the (S. fluitans III dominated) GASB”
4) I wonder is the aim of the study is completed, as I do not find any hypotheses on the origin of Sargassum in the northwestern coast of Cuba. From which part of the Atlantic Ocean are originated beached Sargassum in the northwestern part of Cuba? One or several origins?
Reply: In the discussion (paragraphs 2, 3, and 4) we mention that. Also, we added the following text: “In the winter, the NW Cuban coast likely received sargasso from the (S. natans I dominated) Sargasso Sea, whereas in the summer months, sargasso likely came from the (S. fluitans III dominated) GASB”.
Again, we did not explain this clearly enough; please see above.
5) In the discussion section, it will be interesting to add the article from Ody et al. (2019) which presented also (floating) biomasses estimation more South compared to the study presented here. These (drift) biomasses presented in Ody et al. (2019) are more important compared to the study here (beached biomasses) but comparable with those from Wang et al. (2019) who presented also drift biomasses. Hypotheses could be done also on this decrease of biomasses (from South to North, from drift to beachings).
Reply: We thank the reviewer for this observation, and following the reviewers' recommendation, we have modified the fourth paragraph in the discussion (Lines 165-176) as follows:
During the spring-summer season (April to September), when sargasso in the GASB tends to be more abundant, coasts that commonly do not receive sargasso from the GASB may receive biomass that did not follow the main trajectories of the Caribbean- , Caiman- and Yucatan , Loop Currents, and Gulf Stream (Figure 1). Probably sargasso from the GASB may have reached the NW Cuban coast in the summer of 2020 through similar transport ways. Satellite images did not show important sargasso biomass near Cuba's NW coast during the months when S. fluitans III was the dominant landing morphotype (https://optics.marine.usf.edu/projects/SaWS.html). However, small isolated and scattered rafts are not detected by satellites, and according to Ody et al. (2019), small-size rafts (~30 cm) represented 14% of the rafts observed in situ surveys in the central Tropical North Atlantic in the summer of 2017, and that they were more abundant in the northern part of the eastern Caribbean.
We thank the reviewers for their suggestions, and we hope that you will find this revised version an improvement of the original manuscript.
Sincerely,
Eduardo Gabriel Torres Conde, MSc.
In names of all authors
Ph.D. Student
Unidad Académica de Sistemas Arrecifales-Puerto Morelos, Instituto de Ciencias del Mar y Limnología, Universidad Nacional Autónoma de México
Prolongación Avenida Niños Héroes S/N, Puerto Morelos, Quintana Roo 77580, México.
Emails: etorresconde2@gmail.com
ORCID: 0000-0003-4063-4326
Reviewer 2 Report
The manuscript titled “temporal changes in the composition of beached holopelagic Sargassum spp. along the northwestern coast of Cuba" presents a comprehensive study investigating the distribution, composition, and abundance of holopelagic Sargassum species (S. fluitans and S. natans) in the northwestern tropical Atlantic region. The authors conducted systematic surveys on six beaches along the coast of Cuba, analyzing the morphotype composition of beached Sargassum over a two-year period from June 2019 to June 2021. The study aimed to understand the potential factors influencing the presence of Sargassum, including wind speed, wave height, and specific meteorological conditions.
The discussion delves into the implications of the results and offers valuable insights into the potential influence of wind patterns and ocean currents on Sargassum landings. The comparison of findings with other regions affected by massive Sargassum landings adds depth to the discussion, highlighting the unique characteristics of NW Cuba's Sargassum composition.
Seasonal changes in ocean flow fields and short-term shock events (such as hurricanes) should also be appropriately described. Only by reducing high-frequency variation signals can we effectively evaluate the effects of longer period events (low-frequency factors). The study's findings are insightful and contribute to the ongoing discourse surrounding Sargassum dynamics and their implications in the northwestern tropical Atlantic region. I recommend the manuscript for publication after minor revisions for clarity and consistency.
Author Response
We thank the reviewers for their comments and suggestions. We have made all the suggested changes.
Reviewer #2:
- Seasonal changes in ocean flow fields and short-term shock events (such as hurricanes) should also be appropriately described. Only by reducing high-frequency variation signals can we effectively evaluate the effects of longer period events (low-frequency factors). The study's findings are insightful and contribute to the ongoing discourse surrounding Sargassum dynamics and their implications in the northwestern tropical Atlantic region. I recommend the manuscript for publication after minor revisions for clarity and consistency.
Reply: Thank you very much for your comments; and it worthwhile to follow this suggestion up in future investigations. We do not have information on seasonal changes in ocean flow fields for the study region but now mention in the Materials and Methods section that “During the study period, no hurricanes passed near NW Cuba” (Line 71).
We thank the reviewers for their suggestions, and we hope that you will find this revised version an improvement of the original manuscript.
Sincerely,
Eduardo Gabriel Torres Conde, MSc.
In the names of all authors
Ph.D. Student
Unidad Académica de Sistemas Arrecifales-Puerto Morelos, Instituto de Ciencias del Mar y Limnología, Universidad Nacional Autónoma de México
Prolongación Avenida Niños Héroes S/N, Puerto Morelos, Quintana Roo 77580, México.
Emails: etorresconde2@gmail.com
ORCID: 0000-0003-4063-4326
Reviewer 3 Report
1. Line 66: The study was conducted on six beaches on the NW coast of Cuba. We can see the coordinates for each beach in Table 1, but it is difficult to understand the related location for each beach. So it is better to show a study map which includes these six beaches.
2. Line 76: In Table 1, there is only rocky beach in this study. Why did you choose this beach? What is the effect of rocky beach on this study?
3. Line 128: The NW Cuba coast receives low quantities of sargasso independently of the season and possible distinct origin source. It is necessary to explain more detail that the mean wet biomass landed on the NW Cuban coast was lower than that reported for other regions. What are the import factors?
4. Line 138: The most dominant morphotype composition of beached sargasso in NW Cuba was S. natans I. But S. fluitans III was most dominant in samples from Florida and the Bahamas from 2019 to 2021. NW Cuba is close to Florida, but the dominant morphotype is different during the same period from 2019 to 2021. Why?
5. Line 149: Consistent landings in the winter season were triggered by winds ≥ 10 km/h and wind and wave directions. Which specie is most affected or sensitive by wind and wave in this study?
6. Line 172: “Climate change is causing anomalous temperatures, heatwaves, and abnormal meteorological conditions while species/morphotypes of sargasso exhibit varying thermal tolerances and growth rates.” This is important point. However, what is the effect of varying morphotypes of sargasso on the tidal ecosystem?
Author Response
We thank the reviewers for their comments and suggestions. We have made all the suggested changes.
Reviewer #3:
- Line 66: The study was conducted on six beaches on the NW coast of Cuba. We can see the coordinates for each beach in Table 1, but it is difficult to understand the related location for each beach. So it is better to show a study map which includes these six beaches.
Reply: We respectfully disagree with this recommendation, as in this study we wished to address the relative changes of sargasso arriving to the NW coast, and not the spatial variability; therefore, it is more relevant to include a figure showing the ocean currents that could influence the landings of sargasso in NW Cuba (Fig. 1) than one showing the specific location of study sites and thus we decided to keep just their coordinates in Table 1. For the more interested reader we have now included the variability per site in supplementary table 1 (see also reply to the first reviewer).
- Line 76: In Table 1, there is only rocky beach in this study. Why did you choose this beach? What is the effect of rocky beach on this study?
Reply: The supralittoral composition is just a characteristic of the beaches surveyed along the NW Cuban coastline. These beaches were chosen for logistical reasons.
- Line 128: The NW Cuba coast receives low quantities of sargasso independently of the season and possible distinct origin source. It is necessary to explain more detail that the mean wet biomass landed on the NW Cuban coast was lower than that reported for other regions. What are the import factors?
Reply: There are many coasts in the region that do not receive massive quantities of sargasso; which depends on the position of the coasts in relation to the prevailing current and wind directions. We consider that this question has already been answered in the following:
Lines 124-126: “the geographical location of the Floridian and Gulf of Mexico coasts may favour lower landings of sargasso, which likely also applies to the NW Cuba coasts”
5) Line 138: The most dominant morphotype composition of beached sargasso in NW Cuba was S. natans I. But S. fluitans III was most dominant in samples from Florida and the Bahamas from 2019 to 2021. NW Cuba is close to Florida, but the dominant morphotype is different during the same period from 2019 to 2021. Why?
Reply: Most S. fluitans III samples in Florida were found in the spring-summer season (when the biomass of sargasso in the GASB is much higher), as in NW Cuba. According to our results, sargasso landings to NW Cuba are related to the season and meteorological/oceanographic conditions:
1) In the autumn-winter season, the origin could be the Sargasso Sea (where S. natans I is dominant). Specialized literature stated that the movement of sargasso from the Sargasso Sea to the study area occurs with winds having velocities of winds ≥ 10 km/h and wind and wave directions from the N, E, and ENE, associated with the presence of cold fronts and the influence of Continental Anticyclone (See Lines 147-152).
2) In the spring-summer season, the origin could be from the GASB (where S. fluitans III dominant) by the trajectories of the Caribbean Current, Loop Current, and Gulf Stream or by the Antilles Current and trade winds (See Lines 161-165).
- Line 149: Consistent landings in the winter season when winds had velocities of ≥ 10 km/h and wind and wave directions. Which specie is most affected or sensitive by wind and wave in this study?
Reply: To our knowledge, no studies have been conducted on the differential effect of wind and waves on sargasso species and morphotypes, nor was this the purpose of our study. The morphotype composition depended on the season and the origin of the sargasso (Sargasso Sea vs GASB)
- Line 172: “Climate change is causing anomalous temperatures, heatwaves, and abnormal meteorological conditions while species/morphotypes of sargasso exhibit varying thermal tolerances and growth rates.” This is important point. However, what is the effect of varying morphotypes of sargasso on the tidal ecosystem?
We thank the reviewers for their suggestions, and we hope that you will find this revised version an improvement of the original manuscript.
Sincerely,
Eduardo Gabriel Torres Conde, MSc.
In names of all authors
Ph.D. Student
Unidad Académica de Sistemas Arrecifales-Puerto Morelos, Instituto de Ciencias del Mar y Limnología, Universidad Nacional Autónoma de México
Prolongación Avenida Niños Héroes S/N, Puerto Morelos, Quintana Roo 77580, México.
Emails: etorresconde2@gmail.com
ORCID: 0000-0003-4063-4326